# Advantages of the Incorporation of Luffa-Based Activated Carbon to Titania for Improving the Removal of Methylene Blue from Aqueous Solution

Souad Boumad [1,2,3,4], Antonia Infantes-Molina [2], Isabel Barroso-Martín [2], Elisa Moretti [3], Enrique Rodríguez-Castellón [2,*], María del Carmen Román-Martínez [4], María Ángeles Lillo-Ródenas [4] and Naima Bouchenafa-Saib [1,*]

[1]   Laboratoire de Chimie Physique des Interfaces des Matériaux Appliquées à l'Environnement, Faculté de Technologie, Université Blida 1, B.P. 270 Route de Soumaa, Blida 09000, Algeria; boumadsouad@univ-blida.dz

[2]   Departamento de Química Inorgánica, Cristalografía y Mineralogía (Unidad Asociada al ICP-CSIC), Facultad de Ciencias, Campus de Teatinos, Universidad de Málaga, 29071 Málaga, Spain; ainfantes@uma.es (A.I.-M.); isabel.barroso@uma.es (I.B.-M.)

[3]   Department of Molecular Sciences and Nanosystems, Ca' Foscari University of Venice, Via Torino 155, 30172 Venezia Mestre, Italy; elisa.moretti@unive.it

[4]   MCMA Group, Department of Inorganic Chemistry, Materials Institute (IUMA), University of Alicante, Ap. 99, 03080 Alicante, Spain; mcroman@ua.es (M.d.C.R.-M.); mlillo@ua.es (M.Á.L.-R.)

*   Correspondence: castellon@uma.es (E.R.-C.); saib-bouchenafa@univ-blida.dz (N.B.-S.); Tel.: +34-9521-31873 (E.R.-C.); +213-555-487-804 (N.B.-S.)

**Abstract:** This research aims to study the possible improvement of methylene blue (MB) removal from aqueous solution by hybrid adsorbent-catalysts (*AdsCats*) prepared through the incorporation of activated carbon derived from *Luffa cylindrica* fibers (LAC) to TiO$_2$ photocatalysts. LAC with a specific surface area of 1170 m$^2$/g was prepared by chemical activation with phosphoric acid at 500 °C. TiO$_2$/LAC composites with 70 and 90 wt.% Degussa P25 titania content were prepared. The materials were characterized by N$_2$ physical adsorption, XRD, FTIR, and XPS. The *AdsCats* displayed a very good dispersion of TiO$_2$ over LAC, a surface area of close to 200 or 400 m$^2$/g, depending on the composition, and high crystallinity, showing the presence of anatase and rutile phases. MB removal was studied in two different scenarios: under UV-light after reaching adsorption equilibrium, and under UV-light once the liquid effluent and the *AdsCats* were in contact. The MB removal by LAC has proved to be very efficient, highlighting the predominant role of adsorption over photodegradation. The prepared *AdsCats* have also been compared with their components. The results showed that TiLAC hybrids have superior photocatalytic performance than P25, showing TiLAC-7/3 90% MB removal with respect to the initial concentration just after 30 min of UV light irradiation for both studied scenarios.

**Keywords:** luffa activated carbon; biomass; TiO$_2$; photodegradation; organic pollutants; MB

## 1. Introduction

Over recent years, industrial development has engendered negative effects on the environment throughout the propagation of wastewater containing organic pollutants, especially from textile industries [1,2]. These released wastewaters are charged with different recalcitrant dyes, such as malachite green, rhodamine-B, methylene blue, congo red, etc. Due to their high toxicity, these compounds represent a serious danger both for human health (cancer diseases) and the environment, especially for aquatic life [3,4]. Moreover, they present complex aromatic structures that are hard to eliminate or degrade to less or non-toxic materials by conventional methods, such as flocculation and coagulation [5]. In general, the concentration of these recalcitrant dyes in wastewaters typically ranges from 10 to 200 mg·L$^{-1}$.

Various methods can be used for the removal of pollutants from water, being adsorption and advanced oxidation processes among those of highest effectivity. Adsorption is one of the preferred methods for the removal of dyes from wastewaters [6,7]. The porous structure of the adsorbents allows trapping of organic and inorganic pollutant molecules [8], but once the adsorbent is saturated, a regeneration step is needed for further use. Heterogeneous photocatalysis, based on advanced oxidation, has attracted the researchers' interest for water treatment applications because it is able to degrade organic contaminants to nontoxic compounds [9,10]. A photocatalyst is activated by light of the appropriate wavelength [11–13], and electron-hole pairs able to drive red-ox reactions are created. It is considered a versatile, low-cost and environmentally friendly clean technology [14], able to degrade organic pollutants in both gaseous and liquid phases.

Many metal oxide semiconductors, such as $ZnO$, $CeO_2$, $ZrO_2$, and $TiO_2$, have been used as photocatalysts. Among them, $TiO_2$ is considered to show the best photocatalytic performance and maximum quantum yields; also, it is biologically and chemically inert, and a non-toxic low-cost material [15]. However, it presents a large bandgap energy (3.0–3.2 eV), for which UV irradiation is required [16,17]. Powder $TiO_2$ in a liquid phase photocatalytic process involves the drawbacks of a troublesome separation from the suspension and aggregation, mainly when present at high concentrations. Considering these issues, the preparation of low-cost composite materials easy to recover by sedimentation, and with improved photocatalytic activity (in comparison to available commercial $TiO_2$ photocatalysts) is challenging and attracts many researchers' interest.

Different materials and compounds have been incorporated to $TiO_2$ with the purpose of improving its photocatalytic activity. Among them, silica [18,19], noble metals [20], diatomite [21] and activated carbon [22,23] can be highlighted. The latter is gaining attention due to several advantages, mostly for being low cost, non-toxic, and its affinity for the adsorption of most organic molecules [24]. Its porosity is tailored as desired with the suitable combination of experimental conditions, fitting most coveted applications. Nevertheless, the exploited raw materials (residues or by-products) are biodegradable and available with high content of cellulose [25,26]. Using these precursors encourages both valorization and recycling of wastes, which helps to reduce environmental issues. Thus, preparing and using activated carbon is beneficial from both an economic and environmental point of view. In recent years the use of carbon materials, especially activated carbons (ACs), as supports/additives for $TiO_2$ particles is being widely investigated [27–30]. When studying photocatalytic degradation of pollutants in liquid media reactions, the AC textural properties are very interesting. The characteristic porous structure promotes it to be a suitable support for the titania particles, as its developed surface area permits a good dispersion of titania particles on the surface leading to a better catalyst/pollutant interaction [31]. To increase the contact between the catalyst and the target compounds, activated carbon performs as a co-adsorbent able to transfer the pollutants to the $TiO_2$ surface [15,32]. Moreover, activated carbon can confer higher thermal stability and hinder crystalline phase transformation upon heat treatment, [11]. In addition, it is worth noting the ease of recovery of the photocatalysts by simple filtration or sedimentation from the reaction solution in comparison with the colloidal $TiO_2$ nanoparticles.

In the present study, an activated carbon derived from a scarcely exploited biomass residue, *Luffa cylindrica* fibers, has been prepared and used as an *additive* to commercial titanium dioxide, P25. The purpose was to investigate its performance on the removal of a model cationic dye, methylene blue (MB), from the solution and also, to find out if the incorporation of this activated carbon, named LAC, could lead to *adsorbent-catalyst* composite materials with improved MB removal from solution, especially by photocatalytic degradation. Two $TiO_2$/LAC samples with different compositions have been prepared by mechanical mixing in a mortar and have been thoroughly characterized.

The composite materials have been evaluated under two different scenarios: MB photodegradation under UV-light after some adsorption (dark) period or, directly, MB removal under UV-light once the liquid effluent and the *AdsCats* are in contact. Attention is paid to the relative importance of adsorption and photodegradation.



## 2. Materials and Methods

### 2.1. Materials and Reagents

*Luffa cylindrica* fibers have been collected from the Metidja region in Algeria and used without further purification. A commercial titania, P25 from Degussa, was used. Phosphoric acid (85 wt. % solution), and methylene blue (MB) 3,7-bis(dimethylamino) phenazathionium chloride, $C_{16}H_{18}ClN_3S$ $xH_2O \geq 97.0\%$, were purchased from Sigma Aldrich.

### 2.2. Synthesis of TiO₂/LAC Photocatalysts

Luffa activated carbon (LAC) was prepared as follows: first, the collected fibers were washed several times with hot distilled water to remove dust and organic impurities and dried in an oven overnight at 90 °C. Then, the dried fibers were cut and milled using a grinder. Afterwards, they were sieved to collect samples of the desired size, 800 μm. These samples were put in contact with an 85 wt. % $H_3PO_4$ solution in wt. ratio 3:1 ($H_3PO_4$:luffa fibers) at 80 °C, 3 h under continuous stirring. In this step, $H_3PO_4$ is the activating agent and the main responsible for developing the porous structure of the activated carbon. The obtained mixture was next treated at 500 °C for 1 hour in a tubular furnace with a heating rate of 5 °C/min under 60 ml/min $N_2$ flow [33]. The TiO₂/LAC composites were prepared by a simple mixing method in which Degussa P25 TiO₂ was well blended in a mortar with a certain amount of LAC to obtain mixtures containing 70 and 90 wt.% of P25. The carbon content must be limited to allow the proper light absorption by the active TiO₂. The samples were referred to as TiLAC-x/y, where x/y refer to Ti/LAC weight proportion.

The composite TiLAC-9/1 has been treated at 250, 300, and 350 °C, in air flow, to evaluate any possible influence of the oxidation state of titania and/or the generation of oxygen surface groups on the photocatalytic removal of MB.

### 2.3. Characterization

The evaluation of the textural properties of the materials was conducted by $N_2$ adsorption-desorption at −196 °C and $CO_2$ adsorption at 0 °C in a Quantachrome Autosorb-6B apparatus. The samples were degassed at 250 °C for 4 h. The specific surface area ($S_{BET}$) and total micropore volume ($V_{DR}N_2$) were determined by applying, respectively, the Brunauer–Emmett–Teller and Dubinin–Radushkevich equations to the $N_2$ adsorption data [34]. The mesopore volume ($V_{MESO}$) was estimated as the difference between the volume of $N_2$ adsorbed at P/P₀ = 0.9 and P/P₀ = 0.2, expressed as a liquid [34]. The total pore volume ($V_{Total}$) was determined from the volume of nitrogen adsorbed at a P/P₀ = 0.99 relative pressure and the narrow micropore volume ($V_{DR}CO_2$) was determined by applying the Dubinin–Radushkevich equation to the $CO_2$ adsorption data [34]. The morphology of the LAC sample was characterized using an FEI Quanta 650 FEG-ESEM scanning electron microscopy with SE detector, while the composites were characterized using the Helios NanoLab 650 Dual Microscope model from FEI with an Elstar XHR immersion lens FESEM column and carbon discs as support for the powders.

Powder X-ray diffraction analyses were conducted to identify the crystallinity and phase structure of the TiLACx/y photocatalysts (SEIFERT 2002, Cu Kα (1.5406 Å) radiation, 2θ range from 2° to 80°, scanning velocity 2°/min).

FT-IR spectra of solid samples were registered in the mid-infrared and the close range (Bruker Vertex70, with a Golden Gate Single Reflection Diamond ATR System accessory). The spectral range was 4000–500 $cm^{-1}$ with 4 $cm^{-1}$ resolution and 64 accumulations for spectrum acquisition.

Diffuse reflectance UV-vis (DRUV-vis) spectra were collected with an Agilent Cary 7000 UV-Vis-NIR spectrophotometer equipped with an integrating sphere accessory, in the wavelength range from 200 to 800 nm. The absorption coefficient (α) was calculated as: $\alpha = \ln(1/T)/d$, where T is the measured transmittance and d is the optical path length. Band gap energy, Eg, was determined through the α value ($m^{-1}$) from a plot of $(\alpha h\nu)^{1/2}$ versus photon energy (hν), where h is Planck's constant and ν is the frequency ($s^{-1}$). The intercept of the tangent to the absorption curves was used to estimate the band gap (Eg) value.

The samples were also characterized by X-ray photoelectron spectroscopy (XPS), using the Physical Electronics PHI 5700 spectrometer with non-monochromatic Mg Kα radiation (300 W, 15 kV, 12536 eV) and with a multichannel detector. Binding energy (BE) values were referenced to the C 1$s$ peak (284.8 eV) from the adventitious contamination layer. The PHI ACCESS ESCA-V6.0 F software package and Multipack v8.2b were used for acquisition and data analysis, respectively. A Shirley-type background was subtracted from the signals. Recorded spectra were fitted using Gauss–Lorentz curves in order to determine the binding energy of the different element core levels more accurately. The error in BE was estimated to be ca. 0.1 eV.

*2.4. Methylene Blue Removal by Adsorption and Photocatalytic Degradation*

Methylene blue is the model dye compound used to evaluate the photocatalytic activity of the synthesized composite materials under UV irradiation following ISO procedure for photocatalytic characterization [35]. According to previous experiments [19,36], the initial MB concentration was $6.0 \times 10^{-5}$ M, the photocatalyst dosage was 1.25 g·L$^{-1}$, and the temperature was fixed at 20 °C using a thermostatic bath. The following two types of experiments were carried out: (a) the mixture MB solution and photocatalyst was maintained under stirring in the dark for 30 min to achieve the adsorption equilibrium before starting the irradiation, and (b) irradiation was started just after preparing the mixture that was kept under magnetic stirring. In both cases, a 100 mL Pyrex photochemical reactor with a 125 W high-pressure mercury lamp, operating between 180 and 420 nm with a peak at 366 nm, was used. The photon flux was measured by using a Delta OHM radiometer HD2302.0 leaned against the external wall of the photoreactor containing only pure water. After switching on the lamp, 2 mL aliquots of the aqueous suspension were collected from the reactor and filtered through a 0.45 μm PTFE Millipore disc to remove the catalyst powder.

A Shimadzu UV-2450 UV/V spectrometer was used to determine the dye concentration after calibration, by measuring the absorbance at 660 nm. Since the degradation pathway for the selected dye is known with high reliability [37], the eventual formation of byproducts was checked, monitoring the overall UV–vis spectrum of the solutions recovered at different times during the degradation experiments.

The rate constant $k$ was calculated according to the following Equation (1):

$$\ln \frac{C}{C_0} = -kt \qquad (1)$$

where C is concentration after time t, $C_0$ represents the initial concentration and k is the pseudo first-order rate constant (min$^{-1}$), calculated from the slope ($-k/2.303$) of the MB concentration and time curve (log–linear scale) as follows (2):

$$k = 2.303 \times \text{slope} \qquad (2)$$

## 3. Results

### 3.1. Characterization of the Materials

3.1.1. Textural Properties

Figure 1a represents adsorption–desorption isotherms of N$_2$ at $-196$ °C of the LAC and TiLACx/y samples. According to the IUPAC classification, all the isotherms are type IV with H1 hysteresis loops, indicative of a mesoporous structure [38].

The pore size distribution of the samples depicted in Figure 1b was determined by applying the two-dimensional non-local density functional theory (2D)-NLDFT for a heterogeneous surface to the nitrogen adsorption data. It points out that all samples present a mesoporous structure, with an average pore width of 2–5 nm. This is in good agreement with chemical activation by phosphoric acid, which develops microporous to mesoporous structures [31].

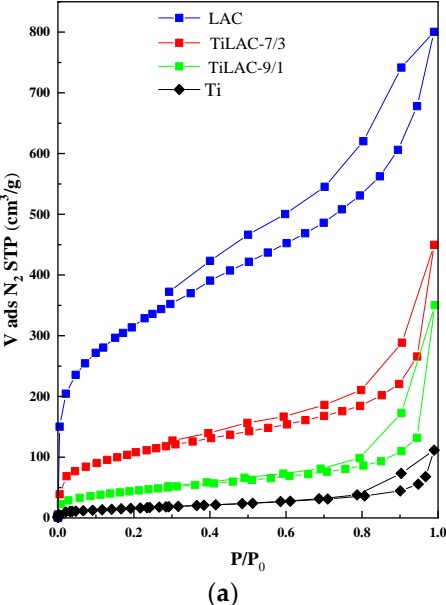
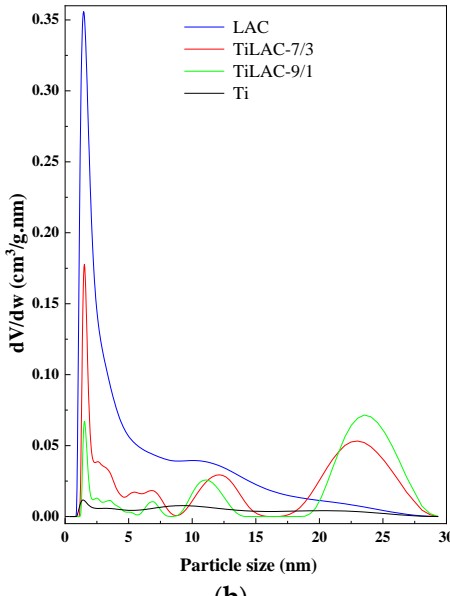

**Figure 1.** (**a**) Adsorption–desorption isotherms of $N_2$ at $-196$ °C and; (**b**) pore size distribution of the prepared luffa activated carbon and TiLAC composites.

Table 1 compiles the calculated apparent BET surface area, the total micropore volume ($V_{DR}N_2$), an estimation of the mesopore volumes from the $N_2$ adsorption isotherms ($V_{MESO}$), the narrow micropore volume ($V_{DR}CO_2$) and the total pore volume ($V_{Total}$). According to Table 1, the composite materials present textural properties in agreement with their components proportion. Although their surface area and porosity are lower than those of LAC, they are noticeably higher than those of the commercial benchmark, P25. In addition to the developed volume of mesopores, the samples also have a perceptible micropore structure [16].

**Table 1.** Textural parameters for P25 $TiO_2$, the prepared luffa activated carbon (LAC) and TiLAC composites, obtained by $N_2$ adsorption–desorption at $-196$ °C and $CO_2$ adsorption at 0 °C.

| Sample | $S_{BET}$ $(m^2 \ g^{-1})$ | $V_{DR} \ N_2$ $(cm^3 \ g^{-1})$ | $V_{MESO}$ $(cm^3 \ g^{-1})$ | $V_{DR} \ CO_2$ $(cm^3 \ g^{-1})$ | $V_{Total}$ $(cm^3 \ g^{-1})$ |
|---|---|---|---|---|---|
| Ti (P25) | 60 | 0.02 | 0.08 | 0.02 | 0.10 |
| LAC | 1172 | 0.54 | 0.64 | 0.21 | 1.18 |
| TiLAC-7/3 | 378 | 0.16 | 0.28 | 0.07 | 0.43 |
| TiLAC-9/1 | 161 | 0.07 | 0.20 | 0.03 | 0.27 |

### 3.1.2. SEM Analysis

Figure 2 shows SEM images obtained for raw luffa, LAC sample and TiLAC-x/y composites. Figure 2a shows that raw luffa presents a homogeneous appearance, with a rich layer of external lignin around the fibers, as already demonstrated in previous work [39]. In addition, the sponge is slightly fibrous with irregular structure and certain cracks and holes. Figure 2b shows the typical morphology of an activated carbon [40], with the pores created due to the carbonization and activation with phosphoric acid treatments. Figure 2c,d display SEM micrographs for the two prepared composites, from which it can be noticed that both phases are in close contact and $TiO_2$ is highly dispersed, covering most of the activated carbon surface homogenously. Note that the white areas are assigned to Ti particles, while the black and gray parts correspond to carbon. Furthermore, no agglomeration of Ti particles on the LAC surface was observed.

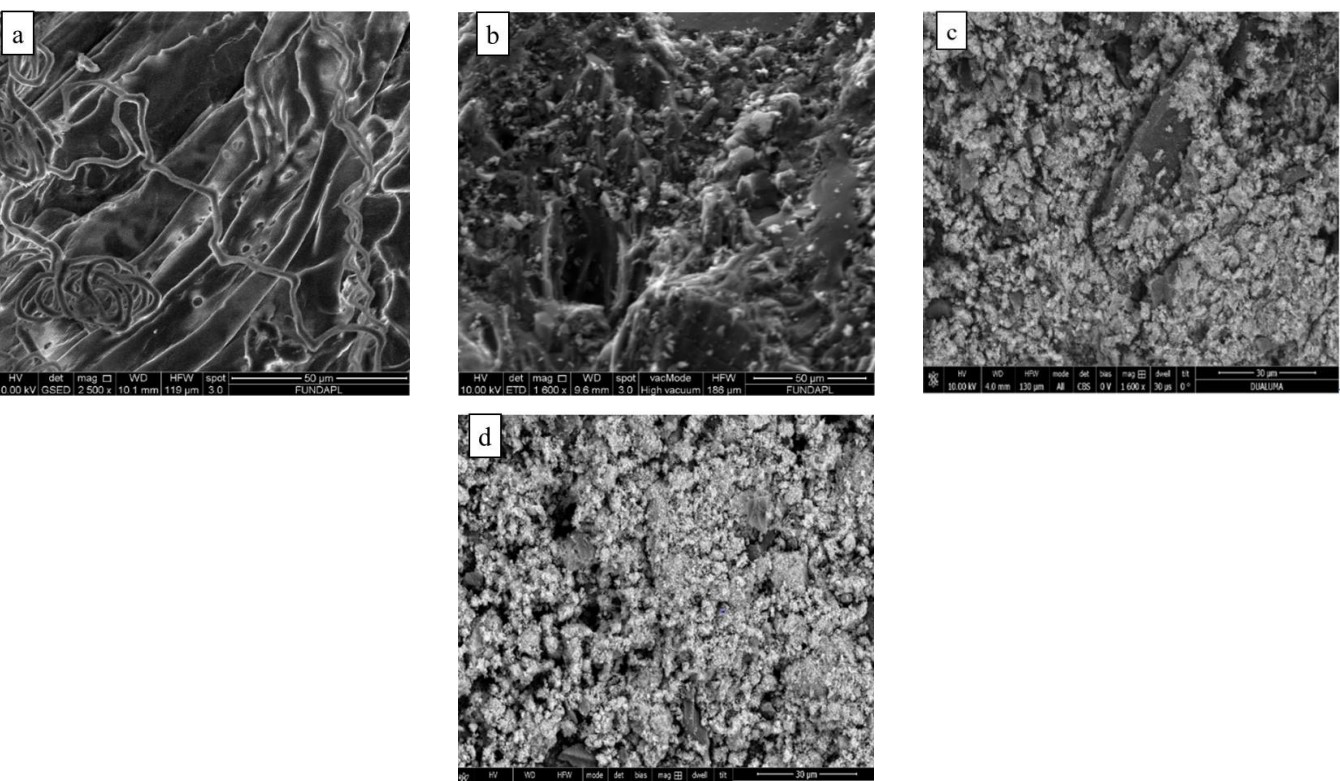

**Figure 2.** Scanning electron microscopy images for (**a**) raw Luffa, (**b**) LAC (**c**) TiLAC-7/3 and (**d**) TiLAC-9/1 (with high vacuum mode (pressure range $10^{-2}$–$10^{-4}$ Pa), imaging resolution of 1.0 nm for 'a' and 'b' and 0.8 for 'c' and 'd' at 10 kV, mode secondary electron image SE, integrated monochromator (UC) and beam deceleration mode. ETD detectors).

### 3.1.3. XRD Patterns

Figure 3 displays XRD diffractograms for pure and composite samples. The XRD pattern of the LAC sample is characteristic of an amorphous activated carbon [16]. Considering the activation temperature (500 °C), graphite formation is not expected. Regarding P25, as expected, the characteristic peaks associated to anatase and rutile are observed ((anatase (*) at $2\theta$ = 25.4° (110), 38.1° (004), 48.2° (200), 54.3° (105), 55.3° (211) and 63.1° (204) and rutile (•) at $2\theta$ = 27.2° (110), 35.8° (101) and 40.9° (111)). For the composite materials, the XRD patterns are similar to that of P25 $TiO_2$, indicating that the incorporation of LAC did not affect the crystallinity of titania.

### 3.1.4. X-ray Photoelectron Spectroscopy

XPS measurements were carried out to study the superficial chemical composition of the samples. Figure 4a shows C 1*s* spectra for LAC and composite samples.

Considering C 1*s* signal, Ti and LAC samples show three contributions at ca. 284.8, 286.5, and 288.5 eV. The main contribution is located at 284.8 eV and assigned to C-C bonds in graphitic carbon and also from the adventitious carbon contamination layer. The other two contributions are due to alcoholic (C-O-H) and carbonyl C=O groups, respectively [41]. Moreover, the LAC sample shows another small contribution at 289.1 eV associated to ester (O-C=O) and $CO_3^{2-}$ groups.

In the case of samples containing both titania and LAC, a new contribution at ca. 283.7 eV appeared. This new contribution is due to the presence of two phases with different conductivity, that is LAC and titania, and is also due to adventitious carbon.

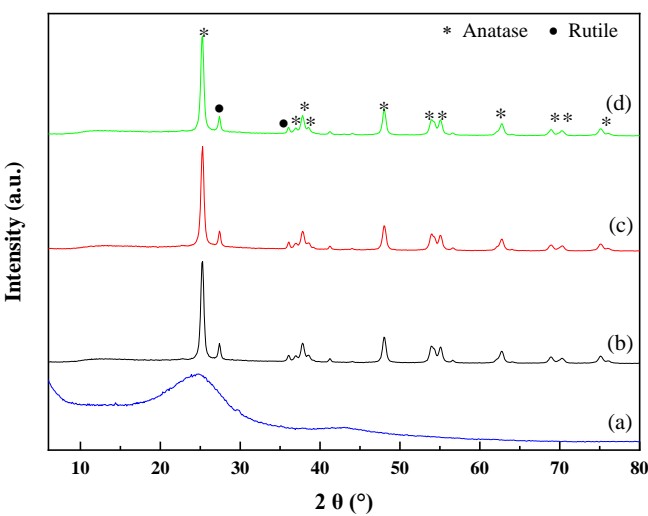

**Figure 3.** X-ray diffractograms of (a) LAC, (b) Ti, (c) TiLAC-7/3 and (d) TiLAC-9/1.

**Figure 4.** Core level spectra for all samples of (**a**) C 1*s*, (**b**) Ti 2*p*, (**c**) O 1*s* and (**d**) P 2*p*.

Ti 2*p* core-level spectra for all samples presented two contributions associated with the doublet Ti 2$p_{1/2}$ y Ti 2$p_{3/2}$, being the latter the most noticeable one at ca. 458.8 eV, which corroborates the presence of Ti$^{4+}$ octahedrally coordinated (see Figure 4b) [42].

Concerning the O 1*s* core-level spectra (Figure 4c), differences between pure samples and composites become more evident. Table 2 includes the corresponding binding energy values. LAC signal presents a major contribution at 533.6 eV, which can be associated with organic C-O bonds in phenolic groups. In addition, a second contribution at lower binding energy, ca. 531.5 eV is present, attributable to carboxylate species. On the other hand, in the pure P25 sample, two signals are observed, one associated with oxygen in the lattice of TiO$_2$ (529.3 eV) and a second one at 532.1 eV, assigned to surface Ti-O-H groups. In the case of composites, signals associated to lattice oxygen in TiO$_2$ and LAC are present.

**Table 2.** BE values corresponding to C 1s and O 1s signals (the relative contribution, in percentage, for the oxygen deconvoluted peaks appears in brackets).

| | BE (eV) C 1*s* Signal | | | | BE (eV) O 1*s* Signal | | | |
|---|---|---|---|---|---|---|---|---|
| LAC | | 284.8 | 286.2 | 288.2 | 290.0 | | 531.6 (33.2) | 533.4 (66.8) |
| TiLAC-7/3 | 283.6 | 284.8 | 286.7 | 288.9 | | 529.7 (82.3) | 531.2 (14.0) | 532.9 (3.7) |
| TiLAC-9/1 | 283.6 | 284.8 | 286.5 | 288.5 | | 529.7 (85.0) | 531.2 (12.3) | 532.8 (2.7) |
| Ti | | 284.9 | 286.6 | 288.7 | | 529.7 (57.1) | 532.0 (42.9) | |

Due to the low presence of phosphorus on the surface, P 2*p* spectra are quite noisy (Figure 4d). Nonetheless, a contribution located at 133.6 eV associated to pentavalent tetra-coordinated phosphorus (PO$_4$), as in phosphate and/or polyphosphate compounds as C-O-PO$_3$, is noticeable, which suggests that P atom is bonded to four oxygen atoms by one double bond and three single bonds, indicating that the P atom in Luffa activated carbon (LAC) is mainly present on the carbon surface by bonding to O atoms resulting from the phosphoric acid (polyphosphate) remaining after the washing step [43–45].

The atomic surface composition obtained from XPS spectra is displayed in Table 3. It can be seen that the atomic composition depends on the content of each material in the sample. A small content of phosphorus has been detected for TiLAC, which can be associated to some traces left behind during the preparation of the LAC activated carbon.

**Table 3.** Atomic surface composition in (%) from XPS spectra.

| | C (%) | O (%) | P (%) | Ti (%) |
|---|---|---|---|---|
| LAC | 93.25 | 6.13 | 0.62 | - |
| TiLAC-7/3 | 38.80 | 43.02 | 0.93 | 17.26 |
| TiLAC-9/1 | 26.92 | 51.34 | 0.37 | 21.36 |

### 3.1.5. FTIR Spectra Analysis

FTIR spectra obtained for P25 and the two TiLAC composites are presented in Figure 5. A medium sharp peak at around 3700 cm$^{-1}$ can be ascribed to the stretching vibration of the O-H bond. The absorption band of water molecules adsorbed from the environment appeared at 1680 cm$^{-1}$ [46]. Two bands at the range 2200–2300 are associated to the carbon dioxide vibration band and medium C=C=C stretching band. A strong absorption peak is observed around 850–1200 cm$^{-1}$ and is attributed to the Ti-O-Ti bond [47]. The wavelength is slightly shifted to a lower wavenumber after the addition of LAC, indicating that the mass of this molecule is reduced, being the frequency of vibration inversely proportional to the mass of a vibrating molecule [48].

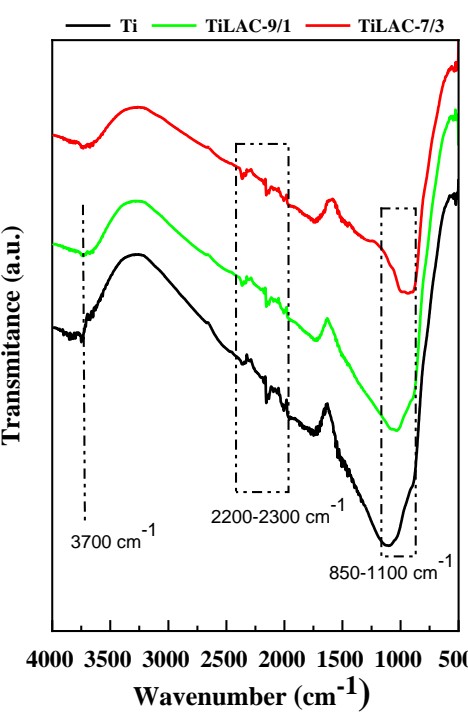

**Figure 5.** FTIR spectra for Ti, TiLAC-7/3, and TiLAC-9/1 samples.

### 3.1.6. UV-Vis Diffuse Reflectance Spectroscopy

The photoresponse of the samples was evaluated by UV-vis diffuse reflectance spectroscopy. The DRUV-Vis spectra and Kubelka–Munk function plots of samples Ti, TiLAC-7/3, and TiLAC-9/1 are displayed in Figure 6. Sample Ti had the lowest absorption in the UV region (<400 nm), while the absorption curves of TiLAC samples presented higher absorption in this region, associated to the intrinsic absorption of $TiO_2$. The presence of activated carbon in the composites strongly affects light absorption, explaining the high intensity of the absorption behavior of these samples in the same region. It is worth mentioning that the DRUV-Vis spectra of the activated carbon is constant and shows a strong absorption in the whole studied wavelength range. This is attributed to the effect of its black color, as reported in many previous studies [49,50]. The band gap energy, $E_g$, was determined through the $\alpha$ value ($m^{-1}$) from a plot of $(\alpha h\nu)^{1/n}$ versus photon energy ($h\nu$), where h is Planck's constant, $\nu$ is the frequency ($s^{-1}$) and the exponent $n$ is the power factor of the optical transition mode. Note that n depends on the nature of the electronic transitions responsible for the absorption and is equal to two for allowed indirect transitions. The intercept of the tangent to the absorption curves was used to estimate the band gap ($E_g$) values of the samples, that were found to decrease according to the increasing amount of activated carbon content: 3.10, 2.85, and 2.66 eV for samples Ti, TiLAC-9/1 and TiLAC-7/3 respectively, indicating an enhancement of $TiO_2$ response in visible light region [51].

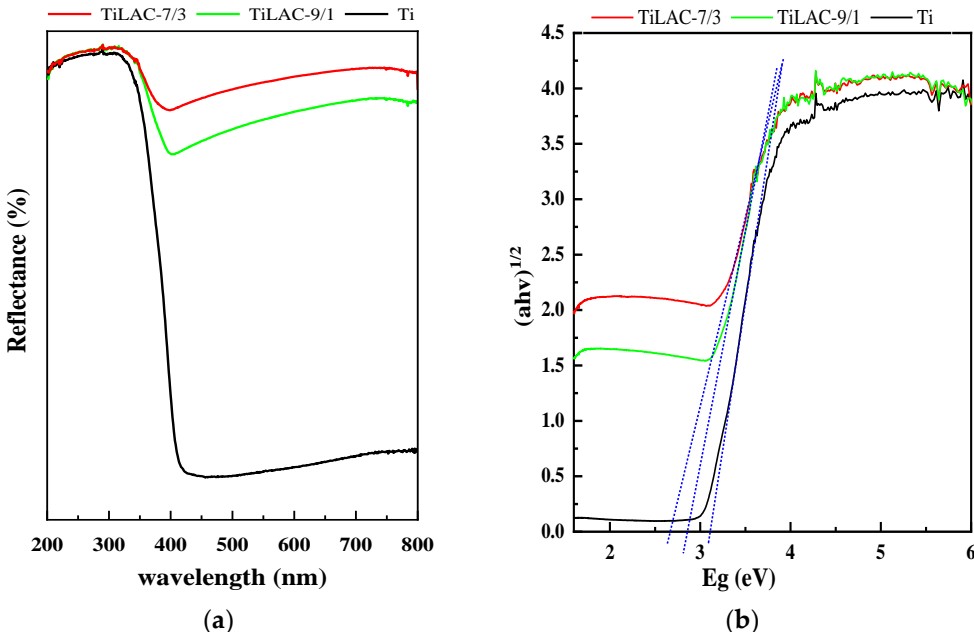

**Figure 6.** (**a**) UV-visible diffuse reflectance spectra and (**b**) energy band-gap vs (ahʋ)$^{1/2}$ of Ti; TiLAC-7/3 and TiLAC-9/1.

### 3.2. Methylene Blue Removal by Adsorption and Photocatalytic Degradation

The prepared TiLAC composites have been tested in the methylene blue dye (MB) photodegradation under UV light irradiation in an aqueous solution. The MB removal from the solution is likely due to a combined adsorption and photodegradation process. Figure 7a shows the variation of $C/C_0$ versus time in the experiment that includes a previous step of 30 minutes in darkness. During this dark step, LAC sample showed a high adsorption capacity of MB, which was expected according to its developed porosity, while the pure titania sample, Ti, showed moderate adsorption, of ca. 10% of the MB initial concentration. However, the TiLAC composites showed much higher MB adsorption than pure Ti, 45% and 74% for TiLAC-9/1 and TiLAC-7/3, respectively, which are in agreement with the higher adsorption capacity of the activated carbon (see Table 1). Under UV-irradiation, the MB removal by the Ti sample reached 94% after 2 h. In contrast, using the TiLAC samples, the MB removal is faster and the total disappearance of the dye from solution occurs within a maximum of 40 min or 60 min, depending on the TiLAC sample (Figure 7a). It must be considered that, in this case, the dye removal is the result of a combination of adsorption and photodegradation processes. A high concentration of MB is assumed to occur in the interphase Ti-LAC for the composite materials through adsorption, in contrast to what happens on Ti's surface, the adsorbed MB can then be photo-oxidized, leading simultaneously to LAC regeneration. This enhancement of the dye removal is in accordance with the LAC content in each sample, as the removal is faster for sample TiLAC-7/3 compared to TiLAC-9/1 and pure titania Ti, respectively. These results are in good agreement with those reported in the literature for methylene blue photocatalytic degradation where similar results using a carbonaceous TiO$_2$ composite material were obtained [16,46,52]. The UV-vis diffuse reflectance analysis shows that the bandgap of the prepared composites is lower than that of titania, which is attributed to the narrowing effect of the addition of LAC, which means an extension of the absorption of TiO$_2$ to the visible light range [53]. Thus, the prepared *AdsCats* could be photoactive in the visible light region.

According to the results obtained when the UV lamp is switched on without the previous 30 minutes in darkness (Figure 7b), it is assumed that MB removal takes place by simultaneous adsorption and photocatalytic processes. The cationic dye is adsorbed from solution on the LAC porosity and then it reacts with the photoactive sites generated through the excitation of the irradiated titania's nanoparticles, resulting in the total dye elimination after 40 to 60 min UV-irradiation. There are no significant differences between the performance of the catalysts in the two tested scenarios. Thus, the second one, skipping

the adsorption equilibrium step, is more efficient for direct MB removal from the point of view of saving time and energy. Figure 8 shows the experimental MB removal curves obtained for each composite and for its individual components (in the amount in which they are present in the composite) and the calculated MB removal by the composite considering the sum of the contribution of each component. These data suggest a synergetic effect between $TiO_2$ particles and LAC and, in this sense, the "excess" MB experimental removal with respect to the theoretical one would be related to photodegradation enhanced by the presence of LAC.

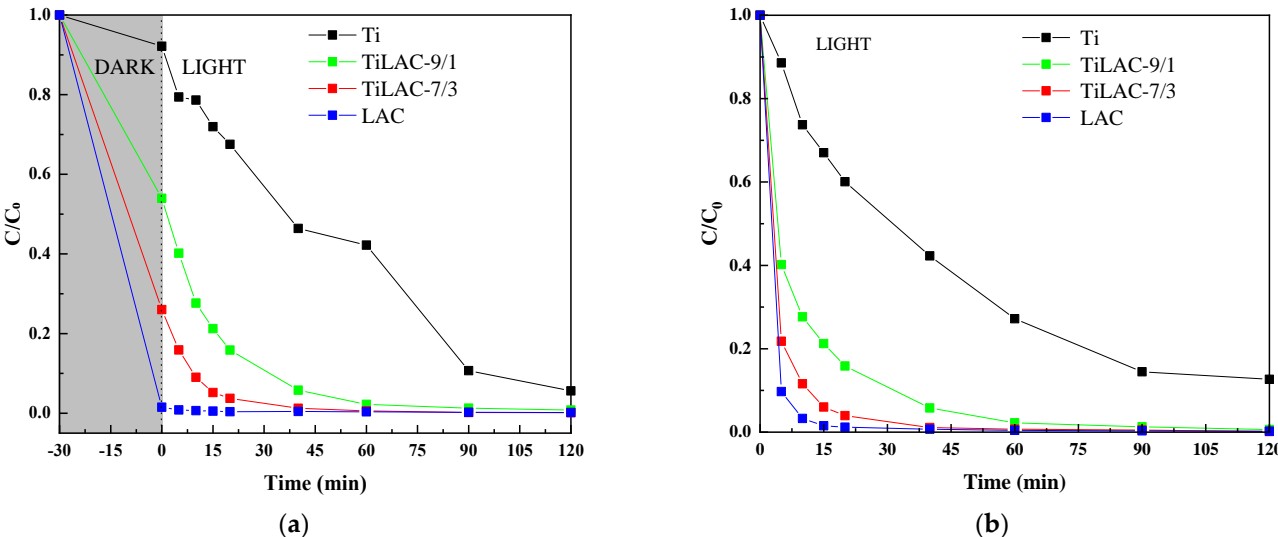

**Figure 7.** Methylene blue removal on Titania P25 (Ti) and TiLAC based catalysts; (**a**) under dark and UV-irradiation; (**b**) under UV-irradiation without the first phase of adsorption equilibrium; with C (MB) = $6.0 \times 10^{-5}$ M, catalyst dosage = 1.25 g·L$^{-1}$ at 25 °C.

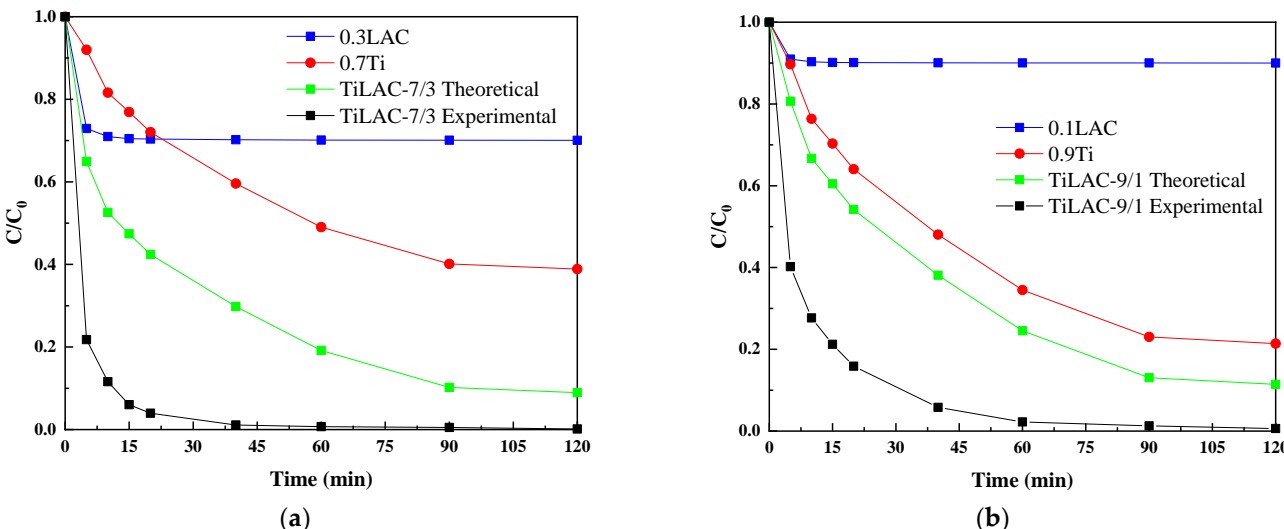

**Figure 8.** Comparison of the experimental MB removal for (**a**) TiLAC-7/3 and; (**b**) TiLAC-9/1 with that predicted according to their individual components and contents.

These results are consistent with the constant rates obtained from the kinetics modeling (Table 4), showing that the photodegradation of methylene blue by all the studied composites follows the first-order kinetics. The rate constant *k* was higher for TiLAC-7/3 and TiLAC-9/1 samples, 0.377 and 0.276 min$^{-1}$ respectively, while the Ti sample yielded the lowest value of 0.182 min$^{-1}$. It can be noticed that the removal efficiencies, previously shown, are in

agreement with the constant rate value and strongly influenced by the content of LAC in each sample, since it seems to ensure a large interfacial surface available between titania particles and the dye molecules, as the BET specific surface areas confirm (see Table 1), providing abundant photoreactive sites and a developed porosity that facilitates the transfer of reactants and degraded products. Therefore, it seems that the higher the contact surface between *AdsCats* particles and MB, the greater the degradation rate.

**Table 4.** The rate constant k values for P25 and the TiLAC composites.

| Sample | $k \ (min^{-1})$ |
| --- | --- |
| TiLAC-7/3 | 0.377 |
| TiLAC-9/1 | 0.276 |
| Ti (P25) | 0.182 |

Additionally, sample TiLAC-9/1 was treated at 250, 300, and 350 °C in a muffle furnace under air in order to investigate the influence of the oxidation state of $Ti^{m+}$ functional species present on the external surface of the composite. The results of the photocatalytic activity experiments are presented in Figure 9. It can be observed that, even though no phase transformations were discerned in the three XRD patterns after thermal treatment (not shown), some significant changes occurred from a catalytic point of view.

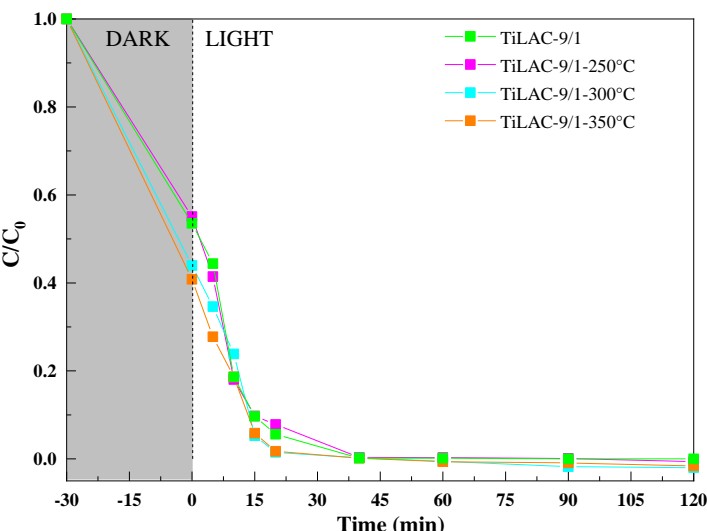

**Figure 9.** Methylene blue removal on TiLAC catalysts, TiLAC-9/1, TiLAC-9/1–250 °C, TiLAC-9/1–300 °C, TiLAC-9/1–350 °C heat treated at different temperatures under dark and with UV-irradiation.

In particular, the thermal treatment at 250 °C did not have any noticeable effect on the photocatalytic activity, while the activities of the samples treated at 300 and 350 °C are higher as the time required for the extensive MB degradation decreases. This improvement can be related to the enhancement in the crystallinity and the formation of larger crystals of titania particles as a result of the thermal effect in the air on the composite materials at temperatures above 300 °C, rather than the oxidation state [54]. Furthermore, heat treatment allowed the adsorption of more oxygen molecules on the surface of the composite by correlating with $Ti^{+4}$ ions. Thus, the presence of these oxygen molecules helps to trap photoexcited electrons and inhibit their recombination with holes. Hence, as the temperature increases, more superoxide anion radicals $O_2^-$ are formed and more hydroxyl radicals are stabilized by preventing combination [55], which is in agreement with Figure 9.

## 4. Conclusions

Composite materials have been prepared by a simple mechanical mixing of $TiO_2$ and an activated carbon derived from *Luffa cylindrica* fibers. The samples, with generic name TiLAC, are identified as *AdsCat* for showing adsorbent and catalytic properties. The effect of carbon content and the thermal treatment temperature in their ability for methylene blue removal from water have been studied. The composites show high BET surface areas related with their LAC content. $TiO_2$ particles are well dispersed on LAC surface, homogenously covering most of the activated carbon surface. The presence of oxygen and phosphorus complexes on the LAC carbon confer high oxidation resistance and active sites that lead to improved adsorption and photocatalytic properties.

The *AdsCat* samples showed a good performance, leading to the total dye removal after 30 min of exposure to the irradiation source. Calculation of theoretical MB removal by the composites based on the addition of each component effect revealed a synergic effect between $TiO_2$ and LAC. The photo-response confirmed that a suitable Eg is obtained for the sample containing 10% LAC (the band gap decreases from 3.10 eV, for pure titania, to 2.66 eV) which is ascribed to the formation of $TiO_2$ LAC heterojunction. This extension of the catalyst absorption to visible light can enhance the catalytic activity. TiLAC photo-catalysts showed, in fact, a higher MB degradation rate than the corresponding bare $TiO_2$. The heat treatment at 300–350 °C for TiLAC-9/1 has a slight influence on its activity by decreasing the time of total degradation up to 15 min.

**Author Contributions:** S.B. prepared the activated carbon and the composites; S.B. and E.M. performed the experiments; all the authors analyzed the data; S.B. wrote the paper. All authors discussed the results and commented on the manuscript. All authors have read and agreed to the published version of the manuscript.

**Funding:** This research was funded by Ministry of Science, Innovation and Universities, projects RTI2018-099668-BC22 and RTI2018-095291-B-I00, Junta de Andalucía, project MA18-FEDERJA-126 and FEDER funds, and GV/FEDER (PROMETEO/2018/076). S.B. acknowledges Mujeres por Africa foundation for the scholarship offer within the 2019 Learn Africa program and, the Ministry of Higher Education and Scientific Research for the 2019–2020 scholarship within the P.N.E. program.

**Institutional Review Board Statement:** Not applicable.

**Informed Consent Statement:** Not applicable.

**Acknowledgments:** The authors thank projects RTI2018-099668-BC22 and RTI2018-095291-B-I00 of Ministry of Science, Innovation and Universities, project UMA18-FEDERJA-126 of Junta de Andalucía and FEDER funds, and GV/FEDER (PROMETEO/2018/076) for financial support. AIM thanks the Ministry of Science, Innovation and Universities for a Ramón y Cajal contract (RYC-2015-17870). Souad BOUMAD thanks Mujeres por Africa foundation for the scholarship offer within the 2019 Learn Africa program and, the Ministry of Higher Education and Scientific Research for the 2019–2020 scholarship of the P.N.E. program.

**Conflicts of Interest:** The authors declare no conflict of interest.

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
