# Peer review of "Advantages of the Incorporation of Luffa-Based Activated Carbon to Titania for Improving the Removal of Methylene Blue from Aqueous Solution"

_applsci, doi:10.3390/app11167607_

Round 1

Reviewer 1 Report

The paper is focused on activated carbon-TiO2 composites with enhanced pollutant removal properties. It is very well written, the background is correctly depicted and the presentation is excellent.

However, there is one aspect that the authors should clarify. In Figure 7, the MB removal by different materials is reported. LAC shows the best performance. The question is: why don't the authors use pure LAC for MB removal then? The answer is a bit obvious, as pure LAC can get saturated and would then need restoration; however, the paper would improve if the authors added a demonstration of the superiority of TiLAC composites over pure LAC for MB removal, or at least addressed this point in the text.

Moreover, there are a couple of minor issues that the authors should address:

  • Why only such high TiO2 loads (70 and 90%) were used?
  • Index letters in Figure 2 should have homogeneous format;
  • Figure 6 should also report the spectrum for pure LAC;
  • Figure 7a and 7b have very different y-axis values, why?
  • In line 329-330 it is stated that "the prepared AdsCats could be photoactive in the visible light region". Can the authors provide proof for this claim e.g., additional MB removal tests performed with Vis irradiation?

Author Response

Reviewer 1:

The paper is focused on activated carbon-TiO2 composites with enhanced pollutant removal properties. It is very well written; the background is correctly depicted and the presentation is excellent.

We thank the Reviewer for the positive evaluation and for all the comments to improve the manuscript. They have all been taken into account in this revised version and detailed answer to each comment is presented as follows.

Q1) However, there is one aspect that the authors should clarify. In Figure 7, the MB removal by different materials is reported. LAC shows the best performance. The question is: why don't the authors use pure LAC for MB removal then? The answer is a bit obvious, as pure LAC can get saturated and would then need restoration; however, the paper would improve if the authors added a demonstration of the superiority of TiLAC composites over pure LAC for MB removal, or at least addressed this point in the text.

A1) The reviewer is totally right and, considering this comment, in the revised MS we have highlighted those aspects. As the reviewer has mentioned, MB removal efficiency for LAC is superior if compared to TiLAC systems, but it is due to adsorption. Thus,  a subsequent LAC regeneration step would be required. In contrast, in the composite photocatalysts, MB removal would also occur by means of a photo-oxidation process that can be enhanced by its adsorption on LAC (a higher concentration of MB in the interphase Ti-LAC is assumed to take place). The adsorbed MB can be photo-oxidized, leading to LAC regeneration. This will be the object of study in a subsequent research work. Furthermore, the present study gives the priority to the enhancement of the TiO2 photocatalytic properties by supporting it on LAC, and the intention is pointed towards a total removal/mineralization of MB rather than adsorption.   

Considering the Reviewer’s comment, this point has been further clarified in the revised version of the MS.

Q2) Why only such high TiO2 loads (70 and 90%) were used?

A1) Thank you very much for this interesting comment. Considering it, we understand that it was not clear enough in the previous submission. On one hand, TiO2 is the photocatalytically active species and, hence, it should be the main constituent. Also, if a large amount of carbon is incorporated to obtain the hybrid photocatalyst, light absorption by the active TiO2 would be strongly decreased. Thus, in previous studies published in the research group (i.e., M.A. Lillo-Ródenas, N. Bouazza, A. Berenguer-Murcia, J.J. Linares-Salinas, P. Soto, A. Linares-Solano, Appl. Catal. B: Environ., 71 (2006), pp. 298-309, https://doi.org/10.1016/j.apcatb.2006.10.004) devoted to hybrid photocatalysts for gas-phase application, catalysts were prepared with 70% TiO2 and 30% carbon, and those percentages were chosen for the reasons previously stated and because, in that particular work, a commercial photocatalyst containing TiO2 and carbon with such composition, around 70% TiO2 and 30% carbon, was used showing very good performances. This point has been clarified in the revised manuscript.

Q3) Index letters in Figure 2 should have homogeneous format;

A3) The index letters have been homogenized. Sorry for this mistake

Q4) Figure 6 should also report the spectrum for pure LAC;

A4) We consider that the absorbance of LAC would remain essentially constant along the wavelength range measured, as it has been reported in the literature (see, as an example, Novel titanium dioxide–graphene–activated carbon ternary nanocomposites with enhanced photocatalytic performance in rhodamine B and tetracycline hydrochloride degradation. July 2017, Journal of Materials Science 52(13). DOI:10.1007/s10853-017-1047-0). Furthermore, It is commonly known that activated carbon (black coloured) exhibits strong adsorption in the whole studied range of wavelengths. It is worth mentioning that, because of the actual situation of the fifth wave of Covid-19 and the partial closure of the universities, the diffuse reflectance analysis of sample LAC could only be performed after two months and, considering the fact that literature has previously shown such constant adsorption along the wavelengths, we have preferred to mention this point in the revised MS for clarity purposes avoiding the performance of the diffuse reflectance analysis of sample LAC, which is what we would have been able to do in a normal period of time.

Q5) Figure 7a and 7b have very different y-axis values, why?

A5) It is just a typo that occurred when representing the graphs by the software (the wrong column was chosen for ‘y’ values as C/C0). The typo has been rectified, sorry for the mistake

Q6) In line 329-330 it is stated that "the prepared AdsCats could be photoactive in the visible light region". Can the authors provide proof for this claim e.g., additional MB removal tests performed with Vis irradiation?

A6) We would like to thank the reviewer for such observation. We are currently performing some tests under simulated solar light which have proved that the AdsCats are indeed photoactive in the visible light region.  

Reviewer 2 Report

Authors have presented the work nicely. So I recommend in the favor of publication of the manuscript. 

Author Response

Thanks for the review's comments. We have revised the English of the manuscript.

Reviewer 3 Report

My comments are attached.

Author Response

Reviewer 3:

The paper describes the preparation and characterization of composites of TiO2 and activated carbon, as well subsequent methylene blue adsorption and photocatalytic degradation. The authors report improved photocatalytic activity attributed to factors such as reduced band gap energy of TiO2 and increased surface area.

However, the controls used in the experiments could be improved, as well as the choice of organic pollutants.

We would like to thank Reviewer 3 for his/her revision and constructive evaluation.

All the comment have been taken into account in this revised version and detailed answer to each comment is presented as follows:

General comments

Introduction: Some aspects of the introduction, such as TiO2 phases and the mechanism of organic degradation, are widely studied and reported. Since this is not new knowledge, try to cut back content on these and include only what is relevant to the current study.

As suggested by the reviewer, this part has been summarized in the revised MS.

Some sentences can be shortened, or split into 2, to make them more comprehensible to readers

The longest sentences have been revised and reconstructed.

Specific comments

Q1) Line 45: What is the significance of the range 10 to 200 mgL-1?

A1) Line 45: It means that in most polluted waters, wastewater discharged during the manufacturing and application processes, these recalcitrant dyes are present in concentrations ranging from 10 to 200 mgL-1.

Q2) Line 46-48: Check the structure of the sentence, as well as grammar

A2) We thank the reviewer for the observation. The sentence has been modified/reconstructed in the revised MS (tracking changes is activated).

Q3) Line 50: “allows trapping of”

A3) Thank you for the correction.

Q4) Line 78-81: “The latter is gaining attention due to its capacity to 80 enhance the photocatalytic performance of titanium oxide”. Most, if not all of the other materials, also serve the function, (enhance the photocatalytic performance of titanium oxide). What makes activated carbon stand out from the other materials?

A4) Several advantages could be highlighted for activated carbons: i) low cost and no-toxicity, ii) affinity for the adsorption of most organic molecules, iii) their porosity can be tailored as desired with the suitable combination of experimental conditions, fitting most desired applications, iv) many residues or by-products can be used as precursors and v) as indicated in references 24 and 25, their use in photocatalysis. Using raw materials encourages both valorization and recycling of wastes which help to reduce environmental issues. Also, the exploited raw materials are biodegradable, abundant, and available worldwide. As a result, preparing and using activated carbon is beneficial both from an economic and environmental point of view.

From a photocatalytic point of view, the characteristic porous structure of activated carbon promotes it to be a suitable support for the titania particles, as its developed surface area permits a good dispersion of titania particles on the surface leading to a better catalyst/pollutant interaction. Also, it is worth mentioning the ease of recovery of the photocatalysts by simple filtration or sedimentation from the reaction solution in comparison with the colloidal TiO2 nanoparticles.

Q5) Line 112: What is the purpose of H3PO4 in this step?

A5) H3PO4 acts as an activating agent and plays a major role in developing the porous structure of the activated carbon. It promotes bond cleavage reactions and the formation of crosslinks via processes such as cyclization, and condensation. It also allows the addition or insertion of phosphate groups that drives a process of dilation that, after removal of the acid, leaves the matrix in an expanded state with a well-developed and accessible pore structure. Furthermore, by using H3PO4, carbons with highly thermally stable phosphorus surface complexes in the forms of C−O−PO3 and C−PO3 are produced. These P groups confer the carbon with high oxidation resistance, surface acidity, and redox sites which improve activated carbon potential for applications in catalysis.

Considering the Reviewer’s comment, in the revised MS we have highlighted that H3PO4 acts as an activating agent and plays a major role in developing the porous structure of the activated carbon.

Q6) Line 164: How was temperature “fixed at 20 °C?

A6) Line 164: The temperature was fixed using a thermostatic bath. The reactor setup had a refrigerating system coupled to a thermostatic bath in which the water circulation maintains the temperature constant at 20 °C. An explanation on that has been included in the revised MS.

Q7) Table 1: Were there repeats to allow for calculation of standard errors?

A7) In general, the accuracy of porosity determination is well-known since not only the samples to characterize but also standards are measured for such purpose. In this case, both considering those standards and the repetition of the characterization of two samples let us conclude that, as expected, the error in porosity estimation is below 5%.

Q8) Line 285-286: Check the sentence for typos

A8) Thanks for this observation, the sentence has been corrected. Now it is: The wavelength is slightly shifted to a lower wavenumber after the addition of LAC, indicating that the mass of this molecule is reduced, BEING the frequency of vibration inversely proportional to the mass of a vibrating molecule [48].

Q9) MB removal experiments: One experiment which can be helpful is the inactivation of MB by photolysis, i.e., in the presence of UV light but with no adsorbent/photocatalyst. The lamp used is powerful enough to cause some photolysis on its own. It will be good to know to what extent this happens before concluding that all the degradation that happens in the presence of light is only due to photocatalysis and adsorption.

A9)

We realize that this point was not totally clear in the previous submission. MB is quite unstable and can be excited under UV irradiation, its photolysis is quite possible under a certain degree of UV irradiation. However, the rate of degradation is not high and only 10 to 20% can be degraded in a large range of time as reported in several studies (Brik, A., Naama, S., Hadjersi, T. et al. Photodegradation of methylene blue under UV and visible light irradiation by Er2O3-coated silicon nanowires as photocatalyst. Reac. Kinet Mech Cat 131, 525–536 (2020); https://doi.org/10.1007/s11144-020-01862-0).

Thus, using photocatalysts is better to improve its removal, decreasing the time-consuming process.

For future studies, I would recommend a more recalcitrant pollutant, since MB is relatively easy to degrade, so the effectiveness of the composites based on MB removal might be inconclusive from a water pollutant removal point of view.

We thank the reviewer for this suggestion, which will be taken into account for future studies.

Round 2

Reviewer 3 Report

I am satisfied with the responses made. However, can you include part of this, in redacted form and/or appropriate references where it appears, in the paper for the benefit of your readers, and to give context on your choice of material:

Several advantages could be highlighted for activated carbons: i) low cost and no-toxicity, ii) affinity for the adsorption of most organic molecules, iii) their porosity can be tailored as desired with the suitable combination of experimental conditions, fitting most desired applications, iv) many residues or by-products can be used as precursors and v) as indicated in references 24 and 25, their use in photocatalysis. Using raw materials encourages both valorization and recycling of wastes which help to reduce environmental issues. Also, the exploited raw materials are biodegradable, abundant, and available worldwide. As a result, preparing and using activated carbon is beneficial both from an economic and environmental point of view.

From a photocatalytic point of view, the characteristic porous structure of activated carbon promotes it to be a suitable support for the titania particles, as its developed surface area permits a good dispersion of titania particles on the surface leading to a better catalyst/pollutant interaction. Also, it is worth mentioning the ease of recovery of the photocatalysts by simple filtration or sedimentation from the reaction solution in comparison with the colloidal TiO2 nanoparticles.

Author Response

Thank you for your suggestion. We have included the required information in the revised manuscript as follows:

Many materials and compounds have been incorporated to TiO2 with the purpose of improving its photocatalytic activity. Among them, silica [18, 19], noble metals [20], diatomite [21] and activated carbon [22, 23] can be highlighted. The latter is gaining attention due to several advantages, mostly for being low cost, non-toxic, and its affinity for the adsorption of most organic molecules [24]. Its porosity is tailored as desired with the suitable combination of experimental conditions, fitting most coveted applications. Nevertheless, the exploited raw materials (residues or by-products) are biodegradable and available with high content of cellulose [25, 26]. Using these precursors encourages both valorization and recycling of wastes which help to reduce environmental issues. Thus, preparing and using activated carbon is beneficial from both an economic and environmental point of view.